# Genome-Wide Association Study of Sweet Potato Storage Root Traits Using GWASpoly, a Gene Dosage-Sensitive Model

**DOI:** 10.3390/ijms252111727

**Published:** 2024-10-31

**Authors:** Robert R. Bowers, Tyler J. Slonecki, Bode A. Olukolu, G. Craig Yencho, Phillip A. Wadl

**Affiliations:** 1United States Department of Agriculture, Agricultural Research Service, United States Vegetable Laboratory, Charleston, SC 29414, USA; robert.bowers@usda.gov; 2Breeding Insight, Cornell University, Ithaca, NY 14853, USA; tjs334@cornell.edu; 3Department of Entomology and Plant Pathology, University of Tennessee, Knoxville, TN 37996, USA; bolukolu@utk.edu; 4Department of Horticultural Science, North Carolina State University, Raleigh, NC 27695, USA; craig_yencho@ncsu.edu

**Keywords:** anthocyanins, beta-carotene, carotenoids, starch, sweet potato, *Ipomoea batatas*, GWAS (genome-wide association study)

## Abstract

Sweet potato (*Ipomoea batatas*) is an important food crop that plays a pivotal role in preserving worldwide food security. Due to its polyploid genome, high heterogeneity, and phenotypic plasticity, sweet potato genetic characterization and breeding is challenging. Genome-wide association studies (GWASs) can provide important resources for breeders to improve breeding efficiency and effectiveness. GWASpoly was used to identify 28 single nucleotide polymorphisms (SNPs), comprising 21 unique genetic loci, associated with sweet potato storage root traits including dry matter (4 loci), subjective flesh color (5 loci), flesh hue angle (3 loci), and subjective skin color and skin hue angle (9 loci), in 384 accessions from the USDA sweet potato germplasm collection. The *I. batatas* ‘Beauregard’ and *I. trifida* reference genomes were utilized to identify candidate genes located within 100 kb from the SNPs that may affect the storage traits of dry matter, flesh color, and skin color. These candidate genes include transcription factors (especially Myb, bHLH, and WRKY family members), metabolite transporters, and metabolic enzymes and associated proteins involved in starch, carotenoid, and anthocyanin synthesis. A greater understanding of the genetic loci underlying sweet potato storage root traits will enable marker-assisted breeding of new varieties with desired traits. This study not only reinforces previous research findings on genes associated with dry matter and β-carotene content but also introduces novel genetic loci linked to these traits as well as other root characteristics.

## 1. Introduction

Sweet potato (*Ipomoea batatas*) is one of the most important food crops worldwide. The ability of sweet potato to grow on marginal lands with minimal inputs, coupled with the large carbohydrate content of the storage roots, makes sweet potato important for global food security [1,2]. In addition to providing an important source of carbohydrates, sweet potato is a good source of several vitamins, especially provitamin A carotenoids, and minerals including iron, calcium, and potassium [3]. The popularity of sweet potato is growing in many parts of the world, including in the U.S., and this is due, in part, to the health benefits associated with their secondary metabolites, especially carotenoids and anthocyanins [3]. Another factor contributing to the increased popularity of sweet potatoes are value-added products including fries, juices, and alcoholic beverages. In addition, sweet potatoes are used as livestock foodstuffs and have potential for biofuel production [4].

Sweet potato belongs to the morning glory family, Convolvulaceae, and the species is highly heterogeneous, outcrossing, frequently incompatible, and has a large hexaploid genome of ~3 Gb (2n = 6x = 90) [5]. This genetic complexity has hindered efforts to characterize the genome of the cultivated sweet potato, but significant progress has been made recently [6]. A large step forward was made with the release of the complete genomes of two closely related, diploid, wild relatives of sweet potato, *I. trifida* and *I. triloba* [7]. These reference genomes enabled GBSpoly, a genotyping-by-sequencing method tailored to polyploids [8,9], GBSapp that is used for SNP copy-number variant calling, and ultra-dense multilocus genetic mapping with MAPpoly [10]. Genome-wide association studies (GWASs) can provide important resources for breeders to improve breeding efficiency and effectiveness by enabling marker-assisted breeding through the identification of genetic loci and genes associated with storage root traits including dry matter, flesh color and skin color.

Globally, there are thousands of sweet potato varieties encompassing incredible genetic diversity that is reflected in phenotypic diversity as evidenced by an array of skin and flesh colors, which range from white or cream, to various lighter yellow and orange shades, to very dark orange and purple colors. Sweet potato varieties with high starch and dry matter content, which are preferred in sub-Saharan Africa, are typically white- or yellow-fleshed and low in β-carotene. Conversely, varieties with orange flesh and high levels of β-carotene are generally lower in starch and dry matter content and are sometimes considered less palatable. The inverse association between starch and β-carotene in sweet potato is well known and there has been a great deal of focus on breaking this genetic linkage to develop more nutritious varieties with high starch content [11,12,13]. Although there is reported to be an inverse association between anthocyanin and starch synthesis in citrus [14], sweet potato varieties with high anthocyanin content also tend to have high starch content (unpublished observation). There is a significant international initiative aimed at boosting the β-carotene content of high-starch crop varieties to enhance their nutritional value and help combat vitamin A deficiency, which affects about one-in-five children in sub-Saharan Africa [15,16]. Elucidation of the genes regulating starch and β-carotene accumulation will support this international effort.

As stated above, there is generally an inverse association between starch and orange pigment accumulation—sweet potato storage root varieties with high starch content are generally unpigmented or yellow, whereas dark orange varieties high in β-carotene have lower starch levels. To increase the nutritional value of sweet potatoes, it is required to develop varieties with increased carotenoid or anthocyanin content that maintain a high starch content. A deeper understanding of the metabolic network controlling the partitioning between starch, carotenoid, and anthocyanin accumulation may enable the development of new sweet potato varieties with increased nutritional value. Here, a genetically diverse set of 384 sweet potato accessions from the USDA germplasm repository were genotyped and a GWAS of the storage root traits of dry matter content, flesh color, flesh hue angle, skin color, and skin hue angle enabled the identification of single nucleotide polymorphisms (SNPs) associated with each of these traits.

## 2. Results

### 2.1. Significant SNPs Associated with Storage Root Traits

The sweet potato storage root phenotypes of percent dry matter (DM), flesh color (FC), flesh hue angle (FHA), skin color (SC), and skin hue angle (SHA) were distributed as shown in Figure 1. GWAS associations were visualized using Manhattan plots, highlighting the regions with significant marker associations with sweet potato root traits. (Figure 2). Twenty-eight SNPs, comprising twenty-one unique genetic loci, were significantly associated with sweet potato storage root traits including the following: dry matter (four loci), subjective flesh color (five loci), flesh hue angle (three loci), and subjective skin color and hue angle (nine loci) (Table 1). For each genetic model tested (additive, 1-dom-ref [1-dominant reference allele], 1-dom-alt [1-dominant alternate allele], 2-dom-ref, 2-dom-alt, 3-dom-ref, and 3-dom-alt), descriptive statistics including variance (R^2^) and *p*-values adjusted for multiple testing with false discovery rate (FDR) corrections as described in the Materials and Methods appear in Appendix A. The rapid decay of linkage disequilibrium (LD) is illustrated in Appendix A, and quantile–quantile (QQ) plots for SNP associations with sweet potato root traits demonstrated a deviation from linear expectation, indicating that non-random associations are present and supporting the reliability of the GWAS findings (Appendix A). As shown in Table 1, each SNP was given a unique name to facilitate discussion. Two of the SNPs associated with SC and SHA are identical (SC2 = SHA1, SC3 = SHA2). In addition, the DM3* SNP comprises two SNPs only twelve base pairs (bps) apart on chromosome 3, the SHA7* SNP comprises four different SNPs within five bps on chromosome 12 at 1.6 Mb, and the SHA8* SNP comprises two different SNPs within three bps on chromosome 12 at 1.9 Gb. In total, 28 SNPs, given 23 unique names, were identified (Table 1). Furthermore, because the FC4 and FC5 SNPs are only about 22 kb apart on chromosome 12 at 22 Mb and the SC2, SHA1 and SC3, SHA2 SNPs are only about 18 kb apart on chromosome 12 at 2.9 Mb, the 23 named SNPs comprise 21 unique genetic loci.

### 2.2. I. trifida and I. batatas Genes Near Root Trait SNPs

A list of all the annotated genes within 100 kb of these SNPs in the *I. trifida* reference genome revealed 511 annotated genes (Appendix A). The 100 kb cutoff was not strictly enforced, in a few cases where gene density is sparse in the *I. trifida* genome, genes that are beyond 100 kb from the SNP were included in the list. Removing genes annotated as “hypothetical proteins” or “conserved hypothetical proteins” as well as redundant gene copies left 365 potential candidate genes in the *I. trifida* reference genome that were associated with the 28 sweet potato storage root trait SNPs identified herein (Appendix A). Due to the complexity of the hexaploid sweet potato genome, it is necessary to use the simpler *I. trifida* and *I. triloba* reference genomes when identifying sweet potato SNPs using GWASpoly. Ultimately, however, our goal is to understand the genetic basis of storage root traits in *I. batatas*, so the Basic Local Alignment Search Tool (BLAST) searches using the SNPs and the immediate surrounding sequences (~30-mers with the SNPs in the middle) were conducted to locate the 28 storage root trait SNPs within the *I. batatas* ‘Beauregard’ reference genome (http://sweetpotato.uga.edu/). The 28 SNPs identified with GWASpoly using the *I. trifida* and *I. triloba* reference genomes, which comprise 21 unique loci, aligned with 102 homologous positions in the *I. batatas* ‘Beauregard’ genome (Appendix A). The number of homologous positions identified in the *I. batatas* ‘Beauregard’ reference genome varied from one to nine positions per SNP, but most SNPs are present in two to six copies (4.4 ± 1.8, mean ± STD), which are located on 4.1 ± 1.4 chromosomes (Appendix A). These results are in general agreement with the observation that the hexaploid sweet potato genome comprises two closely related subgenomes B_1_B_1_B_2_B_2_B_2_B_2_ [18]. The density of genes is greater in *I. batatas*; within the 200 kb regions encompassing the SNPs, on average 28 ± 8 genes are present in the *I. batatas* ‘Beauregard’ genome compared to 22 ± 9 total genes for *I. trifida* (Appendix A). There is also variation in the number of candidate genes present within the 200 kb windows encompassing the SNPs. On average, about twice as many potential candidates were identified in *I. batatas* than *I. trifida*: 30 ± 13 characterized genes compared to 16 ± 7 characterized genes, respectively (Appendix A). This is expected given that there are more copies of the SNPs and a greater gene density in the *I. batatas* genome. In this study, priority is given to the *I. batatas* reference genome in attempts to identify candidate genes associated with the SNPs for the *I. batatas* storage root traits of dry matter and flesh and skin color. In the *I. batatas* ‘Beauregard’ reference genome, 2903 annotated genes were identified within ~100 kb of the SNPs (Appendix A). Removing redundant copies and uncharacterized genes left 629 potential candidate genes in the *I. batatas* reference genome for affecting the storage root traits of dry matter, flesh color and hue angle, and skin color and hue angle (Appendix A). Below, the most likely candidate genes for affecting the SNPs are highlighted. Genes encoding metabolic enzymes involved in starch, carotenoid, or anthocyanin synthesis, transcription factors regulating these genes or other developmental processes, and transporters of metabolites are highlighted.

### 2.3. Candidate Genes Associated with Dry Matter SNPs

Four genetic loci, comprising five SNPs, were associated with dry matter content: chromosome 1 at 27.24 Mb (DM1), chromosome 2 at 23.02 Mb (DM2), 2 SNPs only 12 bp apart on chromosome 3 at 18.88 Mb (DM3*), and chromosome 10 at 3.23 Mb (DM4, Table 1). The DM1 SNP on chromosome 1 is located within an intron of a gene that is annotated as “common central domain of a tyrosinase domain containing protein” in *I. trifida* (itf01g28430) and as “protein of unknown function (DUF_B2219) domain containing protein” in *I. batatas* (Ibat.Brg.01B_G033840.1). A protein BLAST (https://blast.ncbi.nlm.nih.gov/Blast.cgi, accessed on 28 October 2024) identifies this gene as a polyphenol oxidase 1 (*PPO1*). PPOs are generally investigated in the context of food spoilage as the oxidation of phenolic compounds by PPOs causes browning. Though the physiological roles of PPOs are less well characterized, it has been demonstrated that PPOs have important roles in flavonoid metabolism [17,19,20]. Alternate *PPO1* alleles may alter the flux of metabolites through flavonoid pathways (e.g., aurones, phenolic acids, anthocyanins), and thereby affect dry matter indirectly by impacting the availability of metabolites for starch synthesis.

The closest candidate genes to the DM2 SNP on chromosome 2 are a cyclic nucleotide-gated channel gene (Ibat.Brg.02F_G026700.1, 8.5 kb) and a multidrug resistance-associated protein gene (Ibat.Brg.02A_G022320.1, 26.0 kb), which is homologous to ABC transporter C family member 13 transporters that are implicated in transporting glycosylated abscisic acid conjugates into vacuoles [21]. There is also a WRKY DNA-binding protein gene near the DM2 SNP (Ibat.Brg.02B_G028270.1, 71.5 kb). Interestingly, this *WRKY40-related* protein is implicated in transcriptional repression of abscisic acid signaling [22]. Differential expression or activity of structural genes involved in hormone metabolism and transport could affect the balance between the synthesis of starch and carotenoids or flavonoids.

The DM3* SNP on chromosome 3 are in an intron of a gene annotated as “eukaryotic protein of unknown function (DUF914)” (itf03g23340, Ibat.Brg.03A_G026330.1), which appears to be solute carrier family 35 member F1-like gene (nucleotide sugar transporter). The SLC35 gene family of nucleotide sugar transporters are known for importing ADPG into the ER or Golgi, but this transporter could conceivably affect starch metabolism [23]. In addition, there are nearby genes for a H^+^/ATPase (Ibat.Brg.03D_G014090.1, 1.1 kb), two auxin transporter genes (Ibat.Brg.03A_G026350.1, 22.0 kb; Ibat.Brg.03A_G026360.1, 23.0 kb), a Ca^2+/^ATPase (Ibat.Brg.03E_G012950.1, 35.8 kb), and another nucleotide sugar transporter (Ibat.Brg.03A_G026380.1, 45.9 kb) close to DM3*. In summary, it appears that one or more transmembrane transporters underlies the connection between starch metabolism and the DM3* SNPs.

The closest candidate genes to the DM4 SNP on chromosome 10 are an SCP1-like small phosphatase gene (Ibat.Brg.10F_G003480.1, 8.6 kb) and a WRKY DNA-binding protein gene (Ibat.Brg.10C_G004950.1, 9.6 kb), which has homology to *Arabidopsis thaliana WRKY71/EXB1* that plays a key role in initiating axillary meristem development [24]. In addition, starch synthase (*SS4*), which is involved in starch granule initiation, is only 16.8 kb from the DM4 SNP (Ibat.Brg.10A_G004290.1). Further, a bHLH transcription factor (Ibat.Brg.10A_G004370.1, 66.4 kb) which is paralogous to *bHLH144* that, in rice, regulates grain quality by a pathway involving the *NF-YB1-YC12-bHLH144* complex regulating granule-bound starch synthase (*GBSS*) transcription. Finally, there is an HMG-CoA reductase gene (*HMGR*, Ibat.Brg.10D_G005640.1, 69.7 kb) close to the DM4 SNP. HMGR catalyzes the rate-limiting step in the MVA pathway which provides precursors for carotenoid synthesis. Less flux through the MVA could provide more carbon for starch; more flux would leave less carbon for starch and provide more carbon for β-carotene synthesis.

### 2.4. Candidate Genes from Sweet Potato Flesh Color-Associated SNPs

Six SNPs, comprising five genetic loci, were associated with subjective flesh color: chromosome 7 at 5.56 Mb (FC1), five SNPs on chromosome 12 at 3.93 Mb (FC2), 18.96 Mb (FC3), 22.04 Mb (FC4), 22.06 Mb (FC5), and 23.15 Mb (FC6). In addition, three SNPs were associated with flesh hue angle: chromosome 8 at 1.46 Mb (FHA1), chromosome 9 at 23.21 Mb (FHA2), and chromosome 14 at 11.54 Mb (FHA3). The closest characterized gene to the FC1 SNP on chromosome 7 is “protein FLX-like” (itf07g07740; Ibat.Brg.07C_G009680.1, 7.0 kb) and the next closest gene is “LOB-domain containing protein” (Ibat.Brg.07F_G008950.1, 14.6 kb). There is an Myb gene near the FC1 SNP (Ibat.Brg.07A_G006800.1, 73.0 kb) with homology to *Myb62*, which is reported to be a negative regulator of anthocyanin biosynthesis [25]. One or more of these transcriptional regulators may be involved in regulating genes that result in different flesh colors.

The FC2 SNP on chromosome 12 is within an intron of “membrane-anchored ubiquitin-fold protein” (itf12g06370, Ibat.Brg.12A_G007330.1). There are also a chorismate mutase gene (Ibat.Brg.12C_G007840.1, 23.8 kb), an Myb domain protein gene (Ibat.Brg.12C_G007940.1, 64.0 kb) with homology to *Myb16* that is a major regulator of cuticle formation in *Arabidopsis* [26], and a starch synthase gene (granule-bound starch synthase 2, *GBSS2*, Ibat.Brg.12A_G007540.1, 107.6 kb) gene proximate to the FC2 SNP. Chorismate mutase catalyzes the rate-limiting step of the shikimate pathway which creates flavonoid (including anthocyanin) precursors, while *GBSS2* encodes for an enzyme crucial to starch synthesis—the proximity or these two genes near the FC2 SNP creates the potential for a genetic linkage between starch and anthocyanin synthesis.

The FC3 SNP on chromosome 12 is within an intron of a gene annotated as “indoleacetic acid-induced protein” in *I. trifida* (itf12g19540) and as “phytochrome associated protein” in *I. batatas* (Ibat.Brg.12B_G022290.1), both of which appear to be homologs of the auxin-responsive transcription factor *IAA27-like* which has been shown to play a role in adventitious root development in apple and tobacco [27]. Another flesh color candidate gene located near the FC3 SNP is a basic helix-loop-helix (bHLH) DNA-binding superfamily protein gene (Ibat.Brg.12D_G020790.1, 33.0 kb) with homology to *bHLH68* which is implicated in regulating anthocyanin biosynthesis in carrots [28]. One or both transcriptional regulators (*IAA27-like* and *bHLH68*) appear to be the most likely candidate genes for affecting flesh color through the FC3 SNP.

The FC4 SNP on chromosome 12 is within the promoter region of a cytochrome P450, family 714, subfamily A, polypeptide gene (Ibat.Brg.12C_G029120.1), a monooxygenase that deactivates gibberellins [29]. The FC5 SNP is only 20 kb downstream of the FC4 SNP. Other genes close to FC4 and FC5 include: a nucleotide-diphospho-sugar transferase (8.6 kb, Ibat.Brg.12C_G029170.1) and two WRKY DNA-binding proteins, one with homology to *WRKY70* (Ibat.Brg.12F_G025420.1, 25.5 kb), which is activated by salicylic acid and repressed by jasmonic acid [30] and another with homology to *WRKY55* which is also implicated in salicylic acid signaling (Ibat.Brg.12E_G024250.1, 32.6 kb). There is also an Myb domain protein gene (Ibat.Brg.12C_G029230.1, 59.9 kb) with homology to *Myb4* which is a negative regulator of flavonoid synthesis [31]. In addition, a chalcone-flavanone isomerase (*CFI*) family protein gene (78.4 kb, Ibat.Brg.12C_G028940.1), which catalyzes the isomerization of chalcone to naringenin that is a rate-limiting step in anthocyanin biosynthesis, is near the FC4 and FC5 SNPs. Finally, the gene annotated as “protein SPA, chloroplastic” in *I. trifida* (itf12g24270, 60.4 kb) and as “conserved hypothetical protein” in *I. batatas* (Ibat.Brg.12C_G029260.1, 56.8 kb) is the *ORANGE* gene (*OR*) [32]. To date, *OR* is perhaps the best characterized gene for generating orange flesh color due to its impact on β-carotene accumulation in several plant species including cauliflower, melon, and sweet potatoes [32,33,34]. The proximity of *Myb4* and *CFI*, which are key regulators of anthocyanin synthesis, to *OR* may provide a mechanism whereby carotenoid and anthocyanin synthesis are genetically linked.

The closest gene to the FC6 SNP on chromosome 12 is annotated as “phosphate transporter 1.7” (itf12g25870, Ibat.Brg.12A_G029580.1), but a plastid division 1 (*PDV1*) gene is only 2.5 kb away (Ibat.Brg.12E_G025470.1). *PDV1* overexpression was shown to dramatically increase carotenoid accumulation [33]. Other candidate genes close to FC6 include a basic helix-loop-helix (bHLH) DNA-binding superfamily protein gene (Ibat.Brg.12E_G025410.1, 18.8 kb) and a glycogen/starch synthase ADP-glucose type gene (soluble starch synthase 1, *SS1*; Ibat.Brg.12C_G031200.1, 67.2 kb). Note that the locus near FC6 SNP is another location where a gene encoding a key enzyme of starch synthesis *SS1* is physically near a gene *PDV1* implicated in promoting carotenoid accumulation.

The FHA1 SNP on chromosome 8 is in the 3′ UTR of “transcription factor jumonji family protein/zinc finger (C5HC2 type) family protein” (*JMJ*, itf08g02090, Ibat.Brg.08A_G002660.1) and is also just downstream from the coding sequence of another predicted gene that is in the opposite orientation “WWE protein-protein interaction domain protein family” (itf08g02100, Ibat.Brg.08D_G002190.1). *JMJ* is a H3K9 histone demethylase that is demonstrated to play an important chromatin-modifying role in rice flower development [34]. Other close-by candidates are *homeobox-7* (Ibat.Brg.08D_G002170.1, 24.3 kb), which is involved in drought stress and is abscisic acid responsive [35], and a bHLH protein gene (Ibat.Brg.08F_G003550.1, 83.4 kb) with homology to *bHLH94-like*, which is an important developmental regulator [36]. These transcription factors near the FHA 1 SNP may play important regulatory or developmental roles in sweet potato storage root flesh pigment production and accumulation.

The closest gene to the FHA2 SNP on chromosome 9 is annotated as “endonuclease/exonuclease/phosphatase family protein” (Ibat.Brg.09B_G031890.1, 1.4 kb), but an ADP-glucose pyrophosphorylase small subunit gene (*AGPase*, Ibat.Brg.09B_G031940.1) is only 18.1 kb from the FHA2 SNP. *AGPase* catalyzes the first committed step in starch synthesis [37], and starch synthesis and pigment synthesis and accumulation are generally inversely correlated. There is also a glutathione S-transferase theta gene (*GST*, Ibat.Brg.09C_G031750.1, 33.0 kb) near the FHA2 SNP. Multiple lines of evidence implicate GSTs in vacuolar anthocyanin and proanthocyanidin accumulation [38].

The FHA3 SNP on chromosome 14 is in the coding sequence of a gene annotated as “UDP-glycosyltransferase 73B4” in *I. trifida* (itf14g11930) and as “UDP-Glycosyltransferase superfamily protein” in *I. batatas* (Ibat.Brg.14A_G017520.1) the *Arabidopsis* homolog of which has flavonol glycosyltransferase activity [39]. Further research is necessary to determine the role this gene may play in sweet potato flavonoid metabolism.

### 2.5. Candidate Genes from Sweet Potato Skin Color-Associated SNPs

Fourteen SNPs were associated with either subjective skin color or skin hue angle: one SNP on chromosome 6 at 18.71 Mb was associated with subjective skin color alone (SC1), two SNPs on chromosome 12 at 2.82 and 2.84 Mb (only ~20 kb from each other) were associated with both subjective skin color and skin hue angle (SC2, SHA1 and SC3, SHA2), and eleven additional SNPs comprising seven additional loci were associated with skin hue angle (SHA3—SHA8, Table 1). The SHA3 and SHA4 SNPs are on chromosome 1 at 3.49 Mb and 23.12 Mb, respectively. The SHA5 SNP is on chromosome 3 at 5.99 Mb, and the SHA6 SNP is on chromosome 4 at 12.98 Mb. On chromosome 12 there are two SPN clusters, in one five-base pair stretch at 1.57 Mb there are four SNPs (locus analyzed as SHA7*) and at 1.85 Mb there are two SNPs separated by only one base pair (analyzed together as SHA8* locus). The SHA9 SNP is on chromosome 12 at 4.04 Mb.

The closest gene to the SC1 SNP on chromosome 6 is a BTB/POZ domain-containing protein gene (8.6 kb, Ibat.Brg.06A_G016370.1) and other candidate genes near the SC1 SNP include: a zinc ion binding transcription regulator (Ibat.Brg.06A_G016360.1, 15.8 kb), a BSD domain protein (Ibat.Brg.06A_G016330.1, 26.0 kb), and a transducin/WD40 repeat-like protein gene (Ibat.Brg.06A_G016300.1, 55.5 kb). These four genes encode transcriptional regulators that may affect skin color. There is also a copper transporter gene (Ibat.Brg.06A_G016380.1, 19.3 kb) relatively close to the SC1 SNP.

The SC2, SHA1 SNP on chromosome 12 is within an intron of a heavy metal ATPase gene (*HMA1* homolog, itf01g05210, Ibat.Brg.12B_G006270.1) and the SC3, SHA2 SNP on chromosome 12 at 2.84 Mb is only 20 kb downstream of the SC2, SHA1 SNP. Other candidates near these SNPs include several additional membrane transporters: a natural resistance-associated macrophage protein (NRAMP) transporter (Ibat.Brg.12E_G005660.1, 17.9 kb), a nucleotide-diphospho-sugar transferase superfamily protein (Ibat.Brg.12C_G006160.1, 28.7 kb), and an auxin efflux carrier family protein (Ibat.Brg.12F_G003860.1, 49.0 kb). Additional candidate genes near this locus include a 2-oxoglutarate and Fe(II)-dependent oxygenase superfamily protein (Ibat.Brg.12B_G006360.1, 18.3 kb), a bHLH transcription factor (Ibat.Brg.12D_G004200.1, 24.5 kb) that has homology to the bHLH *Spatula* gene that is an important developmental regulator affecting root growth by controlling the size of root meristem in *Arabidopsis* [40].

The SHA3 SNP on chromosome 1 is within the promoter region of a *homeobox-1* gene (itf01g05210; Ibat.Brg.01A_G008010.1) that is homologous to homeobox *RLT1* [41]. The SHA3 SNP is also within 10 kb of an alpha-glucan phosphorylase gene (aka starch phosphorylase, *SP*; Ibat.Brg.01D_G005710.1), and there is an ethylene response factor gene nearby (Ibat.Brg.01E_G003310.1, 12.5 kb).

The SHA4 SNP on chromosome 1 is in the coding sequence of redox responsive transcription factor (itf01g22470, Ibat.Brg.01A_G021900.1), which has homology to ethylene-responsive transcription factor *ERF109-like* [42].

The SHA5 SNP on chromosome 3 is in the 3′ UTR of a gene annotated as “cytochrome c oxidase-related” (itf03g08860, Ibat.Brg.03B_G010360.1). Two genes encoding key metabolic enzymes near the SHA5 SNP are beta-carotene hydroxylase (*BCH*, Ibat.Brg.03B_G011090.1, 52.3 kb) and geranylgeranyl pyrophosphate synthase (*GGPS*) (Ibat.Brg.03B_G010240.1, 74.0 kb). *GGPS* catalyzes a key step in carotenoid synthesis, and expression of sweet potato *GGPS* in *Arabidopsis* increases the carotenoid content [43]. Beta-carotene hydroxylase (*BCH*) oxidizes β-carotene into β-cryptoxanthin and zeaxanthin, and the downregulation of *BCH* increases sweet potato total carotene content including β-carotene [44]. There are also several genes encoding transcriptional regulators near the SHA5 SNP: a WRKY DNA-binding protein encoding gene (Ibat.Brg.03B_G012780.1, 23.9 kb), an Myb gene (Ibat.Brg.03B_G012820.1, 42.5 kb) with homology to *Myb44* which has been implicated in the regulation of starch biosynthesis genes [45], a homeobox-leucine zipper protein family gene (Ibat.Brg.03B_G011210.1, 45.4 kb), and a bHLH DNA-binding superfamily protein gene (Ibat.Brg.03F_G010380.1, 48.6 kb).

The closest gene to the SHA6 SNP on chromosome 4 is annotated as “time for coffee” (itf04g14940; Ibat.Brg.04D_G016900.1). The most compelling candidate gene for affecting skin color near the SHA6 SNP is UDP-glucose:flavonoid 3-o-glucosyltransferase (*UFGT*, Ibat.Brg.04A_G012160.1, 49.2 kb), which catalyzes the final step of anthocyanin biosynthesis.

The SHA7* locus comprising four SNPs on chromosome 12 are in the coding sequence of a gene annotated as glycosyl hydrolase family 38 protein (itf12g02630, Ibat.Brg.12A_G003460.1). Also near this SNP are a basic-leucine zipper (bZIP) transcription factor family protein (Ibat.Brg.12C_G004120.1, 36.8 kb) and a cluster of three genes—a flavanone 3-hydroxylase gene (*F3H*, Ibat.Brg.12C_G004290.1, 68.2 kb), a chalcone-flavanone isomerase family protein gene (*CFI*, Ibat.Brg.12C_G004300.1, 69.2 kb), and a transducin/WD40 repeat-like superfamily protein gene (Ibat.Brg.12C_G004320.1, 73.0 kb), which has homology to *anthesis promoting factor 1* that is a component of chromatin regulatory complex [46]. This three-gene cluster is also present in *I. trifida* (itf12g02720, itf12g02730, itf12g02750) and exists in multiple copies in the *I. batatas* genome. *CFI* and *F3H* encode the first two enzymes of the anthocyanin biosynthetic pathway, *CFI* converts tetra-hydroxychalcone to the flavanone naringenin, and *F3H* hydroxylates flavanones to form dihydroflavonols.

The SHA8* SNPs on chromosome 12 are within the 3′ UTR of a galactinol synthase gene (itf12g03030, Ibat.Brg.12B_G004140.1), and also near SHA8* are a heavy metal transport/detoxification superfamily protein (Ibat.Brg.12F_G002790.1, 9.2 kb), a dihydroflavonol 4-reductase gene (*DFR*, Ibat.Brg.12C_G004710.1, 45.9 kb), and an Myb domain protein (Ibat.Brg.12E_G003940.1, 71.7 kb). *DFR* encodes a key gene of the anthocyanin biosynthetic pathway, that reduces dihydroflavonols to leucoanthocyanidins.

The SHA9 SNP on chromosome 12 is within an intron of a transmembrane amino acid transporter family protein (itf12g06590; Ibat.Brg.12A_G007640.1). The genetic locus +/− 100 kb of the SHA9 SNP on chromosome 12 at 4.04 Mb partially overlaps with the 200 kb window surrounding the FC2 SNP on chromosome 12 at 3.93 Mb. Two genes encoding key metabolic enzymes near SHA9 are a starch synthase gene (granule-bound starch synthase 2, *GBSS2*; Ibat.Brg.12B_G007440.1, 11.9 kb) and a chorismate mutase gene (Ibat.Brg.12D_G007270.1, 96.6 kb). As noted above, both *GBSS2* and chorismate mutase were also associated with the FC2 SNP, although different alleles of these genes are closer to each SNP. There are also several genes encoding transcriptional regulators near SHA9 including two transducin/WD40 repeat-like superfamily proteins (Ibat.Brg.12C_G008110.1, 15.3 kb; Ibat.Brg.12D_G007380.1, 50.8 kb), two Myb family transcription factors (cyclin D binding Myb Transcription Factor 1 [*DMTF1*], Ibat.Brg.12C_G008030.1, 15.9 kb; and Myb 16-like, Ibat.Brg.12B_G007380.1, 48.5 kb), and a homeobox protein (zinc-finger homeodomain protein 9-like; Ibat.Brg.12B_G007490.1, 28.6 kb).

## 3. Discussion

### 3.1. Genetic Regulation of Sweet Potato Storage Root Traits

The genetic underpinnings of sweet potato storage root starch and pigment accumulation are complex and involve the regulated expression and interplay of a large number of genes. These genes can be divided into three classes: (1) transcriptional regulators including transcription factors and histone modifiers, (2) metabolic enzymes and accessory proteins, and (3) metabolite transporters. These gene classes are connected: transcription regulators drive the expression of structural target genes including metabolic enzymes and metabolite transporters. Here, multiple candidate genes for affecting key agronomically relevant sweet potato storage traits are identified, and most of these genes fit into one of these three functional classes as do candidate genes for affecting these traits that were previously identified.

Several studies have investigated the genetic basis of various agronomic traits in sweet potato by quantitative trait loci (QTL) and genome-wide association study (GWAS). Cervantes-Flores et al. 2010 first identified sweet potato QTLs associated with dry matter, starch, and β-carotene content in progeny from a cross between ‘Tanzania’ (African landrace, white-fleshed, high starch) and ‘Beauregard’ (U.S. cultivar, orange-fleshed, low starch) [11]. This early study found about a dozen QTLs associated with both dry matter or starch content and about eight QTLs associated with β-carotene levels. A negative correlation between starch and β-carotene content was observed [11]. These QTLs were mapped onto the AFLP (amplified fragment length polymorphism) map described in Cervantes-Flores et al. 2008 [47], but no specific genes were described. More recently, Haque et al. 2020 [12] used GWAS and QTL analysis to investigate sweet potato dry matter, starch, and β-carotene content in progeny from a cross between the Japanese cultivars ‘J-Red’ (orange-fleshed) and ‘Chosu’ (white-fleshed). This study also noted a negative correlation between starch and β-carotene and identified multiple QTLs on linkage groups 7 and 8, but no specific genes are mentioned [12].

Wu et al. 2018 constructed high-quality reference genomes of two wild, diploid relatives of sweet potato *I. trifida* and *I. triloba*, both of which are very closely related to *I. batatas* [7]. Using these reference genomes and resequencing data from sweet potato landraces used in African breeding programs as well as transcriptomic data from ‘Tanzania’ and ‘Beauregard’, several genes involved in carotenoid synthesis were identified as candidates for determining sweet potato flesh color including phytoene synthase (*PSY,* itf03g05110), phytoene desaturase (*PDS*, itf11g08190), zeta-carotene isomerase (*Z-ISO,* itf04g12320), and lycopene beta-cyclase (*LCB*, itf04g32080) [2]. Zhang et al. 2020 [48] used data from Wu et al. 2018 [7] and Ding et al. 2017 [49] and employed GWAS and expression QTL analysis of transcriptomic data from 104 sweet potato accessions popular in China that have variation in anthocyanin content of storage root flesh. The most significant distant eQTL hotspot associated with flesh color was located on chromosome 12 (20.42 Mb–20.92 Mb), and the most significant GO (gene ontogeny) classifications of the genes in the 200 kb windows encompassing the QTLs were flavonoid biosynthetic process and phenylpropanoid metabolic process [48]. Further, *IbMYB1-2* [50] was proposed to be a master regulator of anthocyanin biosynthesis. In addition, Gemenet et al. 2019 [13] used a biparental mapping population of ‘Beauregard’ (orange-fleshed) and ‘Tanzania’ (white-fleshed) to identify two major QTLs on chromosomes 3 and 12 associated with dry matter and β-carotene. The inverse correlation between starch and β-carotene content was largely attributed to physical linkage between the sucrose synthase (*SuSy*, itf03g05100) and phytoene synthase (*PSY*, itf03g05110) genes on chromosome 3 and a trans effect from the *ORANGE* gene (*OR*, itf012g24270) on chromosome 12 due to its impact on *PSY* and chromoplast development [13].

In our study, GWAS was employed to identify 28 SNPs, comprising 21 unique genetic loci, associated with the storage root traits of dry matter, flesh color, and skin color in a 384-accession subset of the USDA sweet potato germplasm that was constructed previously and largely maintains the genetic diversity of the entire USDA collection [8,9]. Examination of the genetic loci in the *I. trifida* and *I. batatas* reference genomes (http://sweetpotato.uga.edu/) enabled the identification of candidate genes within 100 kb of the SNPs that may regulate the expression of these storage root traits (Appendix A). Genes considered most likely candidates include metabolic enzymes, transmembrane transporters, and transcriptional regulators and are mentioned in the results above and are also listed in Appendix A. The results from this study corroborate and extend the previous studies described above.

### 3.2. Metabolism of Sweet Potato Storage Root Starch, Carotenoids, and Anthocyanins

As shown in the schematic (Figure 3), in the current study of sweet potato storage root dry matter and color we identified *ORANGE* (*OR*) and twelve additional candidate genes that encode key metabolic enzymes involved in the synthesis of starch, carotenoids, or anthocyanins. Two SNPs associated with flesh color in this study are on chromosome 12 about 60 kb and 80 kb from the *OR* gene (Ibat.Brg.12C_G029260.1). This result confirms previous research implicating this locus and gene in flesh color and dry matter in a ‘Beauregard’ × ‘Tanzania’ population [13] and extends this finding to additional *I. batatas* germplasm. Figure 3 illustrates the connections of central carbon metabolism to starch, carotenoid, and anthocyanin biosynthesis. Central glucose metabolism including glycolysis and the tricarboxylic acid cycle (TCA), the shikimate pathway, the MVA (mevalonic acid) pathway, and the MEP (methylerythritol phosphate) pathway are illustrated in an abbreviated manner. In addition, starch and carotenoid biosynthetic pathways, which occur in plastids, and the anthocyanin biosynthetic pathway, which occurs in the cytosol, are illustrated. Enzymes in bold indicate candidate genes associated with SNPs for dry matter or skin or flesh color from this study. In the top left, sucrose from photosynthetic “source” tissue enters the sweet potato root cell “sink”. Sucrose synthase (*SuSy*) can convert sucrose to uridine-5-diphoshate (UDP)-glucose plus fructose (Fruc), which can enter glycolysis. UDP-glucose can be converted to adenosine-5-diphosphate-glucose (ADPG) that can enter the amyloplast (specialized starch-storing plastid) for starch synthesis. As mentioned above, *SuSy* was suggested to be an important genetic determinant of dry matter in a previous study [13]. The canonical starch synthesis pathway, on the other hand, involves ADPG synthesized from glucose 1-phosphate (G1P), derived from glycolytic Glc and Fruc, inside the plastid by ADPG pyrophosphorylase (*AGPase*) the small subunit of which was identified as a candidate associated with flesh hue angle here (Ibat.Brg.09B_G031940.1). Post-transcriptional regulation of *AGPase* plays a key role in starch synthesis [51]. As shown, *AGPase* can produce ADPG from G1P derived from either starch breakdown via starch phosphorylase (*SP*, aka alpha-glucan phosphorylase, Ibat.Brg.01D_G005710.1) within plastids or from cytosolic glycolysis-derived G1P. Soluble starch synthase 1 (*SS1*; Glycogen/starch synthases, ADP-glucose type, Ibat.Brg.12C_G031200.1), soluble starch synthase 4 (*SS4*; “starch synthase” Ibat.Brg.10A_G004290.1), and granule-bound starch synthase 2 (*GBSS2*; “starch synthase”, Ibat.Brg.12B_G007440.1) are all key enzymes of starch synthesis and were all identified here as candidate genes for affecting dry matter and color of sweet potato storage roots. Studies have demonstrated that genetic modification of these starch biosynthetic enzymes alter starch metabolism in sweet potatoes [52,53].

Carotenoids synthesis is illustrated as occurring in specialized plastids termed chromoplasts. The carotenoid precursors, isopentyl pyrophosphate (IPP) and dimethylallyl pyrophosphate (DMAPP), can be derived from the cytosolic MVA pathway, which uses acetyl CoA (Coenzyme A) as a precursor with the rate-limiting step being catalyzed by HMG CoA reductase (*HMGCR*, hydroxy methyl glutaryl CoA reductase, Ibat.Brg.10D_G005640.1), or IPP and DMAPP can be derived from the plastid MEP pathway, which uses the glycolytic intermediates glyceraldehyde 3-phosphate (G3P) and pyruvate as precursors. Geranylgeranyl pyrophosphate synthase (*GGPS*, Ibat.Brg.03B_G010240.1) catalyzes the synthesis of geranylgeranyl pyrophosphate (GGPP) from IPP and DMAPP. GGPP is a precursor of other terpenoids including chlorophylls and gibberellins in addition to carotenoids. The first carotenoid, phytoene, is synthesized by phytoene synthase (*PSY*) which is stabilized by the *OR*, and *OR* is involved in promoting chromoplast development. Plastid division 1 (Ibat.Brg.12E_G025470.1) is another candidate gene from this study that plays a key role in plastid development and has been shown to affect β-carotene levels in other species [33]. Beta-carotene hydroxylase (*BCH*, Ibat.Brg.03B_G011090.1) oxidizes β-carotene thereby decreasing β-carotene levels, and carotenoids produced downstream of *BCH* can be converted to the hormone abscisic acid.

The isopropanoid pathway for anthocyanin biosynthesis is also illustrated. The glycolytic intermediate PEP (phosphoenolpyruvate) and the pentose phosphate pathway (not illustrated) intermediate E4P (erythritol 4-phosphate) can enter the shikimate pathway, which occurs within plastids and forms precursors for anthocyanin synthesis and wherein chorismate mutase (Ibat.Brg.12C_G007840.1; Ibat.Brg.12D_G007270.1) catalyzes the rate-limiting step [54] and phenylalanine (Phe) is synthesized. Phe is converted to the flavonoid precursor p-coumaroyl. Chalcone-flavanone isomerase (*CFI*, Ibat.Brg.12C_G028940.1), flavanone 3-hydroxylase (*F3H*, Ibat.Brg.12C_G004290.1), dihydroflavonol reductase (DFR, Ibat.Brg.12C_G004710.1), and UDP-glucose flavonoid 3-O-glucosyltransferase (*UFGT*, Ibat.Brg.04A_G012160.1) act consecutively in the synthesis of anthocyanins and, remarkably, each of these genes were identified here as potentially important sweet potato root trait modifiers.

### 3.3. Transmembrane Transporters Affect Sweet Potato Metabolism

In Figure 3, the blue icons in the schematic represent the transmembrane transporters. As illustrated, anthocyanins are stored in vacuoles and thus require transmembrane transport. Transmembrane transporters are included to illustrate that metabolites such as ADPG may require transport across membranes, depending on whether ADPG used for starch synthesis is synthesized in the cytosol or plastid. Differences in expression of transporters for ADPG or G1P would be expected to impact metabolic flux. In addition, transporters such as H^+^/ATPases (Ibat.Brg.03D_G014090.1), are involved in establishing electrochemical gradients utilized by other transporters [55]. Any mutations affecting the expression or structure of any of the enzymes or metabolite transporters illustrated in the figure could impact storage root traits of starch and pigment production. Five of the twenty-eight storage root trait SNPs identified here are located within genes for transmembrane transporters, and multiple additional genes encoding metabolite and ion transporters are located close to many of the SNPs—Table 2 lists some of these transporter genes and a complete list of all the transmembrane transporter genes identified here appears in Appendix A.

### 3.4. Transcription Factors Implicated in Expression of Sweet Potato Storage Root Traits

In many instances, it is the regulation of the expression and activity of the metabolic enzymes of biosynthetic pathways, rather than the presence or absence of these structural genes in genomes, that determines the phenotypic outcomes. Several families of transcription factors are known to be involved in regulating the expression of target genes controlling starch, carotenoid, and anthocyanin biosynthesis; these transcription factor families include bHLH (basic helix-loop-helix), Myb (myeloblastosis viral oncogene homolog), WD40 (WD repeats contain histidine-glycine dipeptide at N-terminus of motif and tryptophan-aspartic dipeptide acid at C-terminus), and WRKY (containing a WRKY domain that is a WRKYGQK motif plus an atypical zinc finger). There are many instances where “MBW” complexes, which contain an Myb, bHLH, and a WD40 repeat-containing protein, regulate the synthesis of genes involved in anthocyanin synthesis [58]. In another example, an Myb/bHLH complex is reported to regulate anthocyanin synthesis in the red-centered fruit of the kiwifruit (*Actinidia chinensis*) [59]. Among the list of candidate genes within 100 kb of the SNPs identified here for affecting storage root traits, are six bHLH genes, seven Myb genes, and three WD40-repeat genes that appear to encode transcriptional regulators (the WD40 repeat domain is involved in protein–protein interactions, not exclusive to transcription factors or chromatin modifiers), and five WRKY family transcription factors (Table 3). In addition, four of the thirty-three SNPs are located within genes for transcriptional regulators; a homeobox gene, and a histone methyltransferase, redox responsive, and a phytochrome-associated protein (Table 3).

### 3.5. Sweet Potato Genomic Resources

Although there are several *I. batatas* genome sequences available (sweetpotato.uga.edu, ncbi.nlm.nih.gov/datasets/taxonomy/4120, https://sweetpotato.com/, accessed on 28 October 2024), it is not straightforward to compare these to each other and the extent of gene annotation and curation is still in its infancy. In this study, genomic data were primarily used from the Genomic Tools for Sweetpotato Improvement Project (GT4SP, https://sweetpotatogenomics.cals.ncsu.edu/, accessed on 28 October 2024) and SweetGAINS (Genomic Advances and Innovative Seed Systems https://buell-lab.github.io/sweet_gains.html, accessed on 28 October 2024) that is available on the Sweetpotato Genomics Resource website (http://sweetpotato.uga.edu/, accessed on 28 October 2024). As an illustration of the difficulty in identifying specific genes, a keyword search for “Myb” on the Sweetpotato Genomics Resource website returns 2194 results of predicted proteins from the *I. trifida*, *I. triloba*, and *I. batatas* genomes (accessed 17 May 2024). The vast majority of the >1500 *I. batatas* Myb genes are annotated simply as “myb domain protein”. As another example, four copies of the ORANGE gene were identified in the *I. batatas* ‘Beauregard’ genome herein (this study used genome v1, the I. batatas ‘Beauregard’ v3 genome was released 2/1/2024 during preparation of this manuscript), but each of these genes (Ibat.Brg.12B_G026770.1, Ibat.Brg.12C_G029260.1, Ibat.Brg.12D_G025600.1, and Ibat.Brg.12F_G025620.1) are annotated as “conserved hypothetical protein.” In our attempts to locate candidate genes associated with the sweet potato storage root trait SNPs identified in this study, genes annotated as “hypothetical protein” or “conserved hypothetical protein” were generally excluded for manageability. Additional annotation and curation of the sweet potato genome will aid future efforts towards elucidating genetic mechanisms underlying important agronomic traits.

The ORANGE (*OR*) gene was first identified in cauliflower (*Brassica oleracea* var. *botrytis*) as underlying a gain-of-function, semi-dominant mutation resulting in orange, β-carotene accumulating curds which are unpigmented in plants without the mutation [32]. In addition, the “golden SNP” of the melon (*Cucumis melo*) *OR* gene results from a single amino acid, arginine to histidine, change that dramatically increases β-carotene accumulation in melon fruits [73], and a similar single-amino acid arginine to histidine substitution in *Arabidopsis* as well as an arginine to alanine substitution also dramatically increase the β-carotene accumulation [74]. Further, a similar single amino acid substitution of the sweet potato *OR* gene (R96H) has a similar effect in sweet potatoes [75]. In the *I. batatas* ‘Beauregard’ genome, all four alleles of OR encode 313 aa proteins. The proteins encoded on chromosomes 12B and 12C are identical to each other and to the *I. trifida* sequence (itf12g24270), but on chromosome 12D, there are R34M and S138L mutations and on chromosome 12F, there are H25L and V56L mutations. The sequence of the orange-fleshed *I. batatas* ‘Shinhwangmi’ *OR* (HQ828087) has three single amino acid changes compared to the orange-fleshed ‘Beauregard’ S61A, I124N, E132K (‘Beauregard’:‘Shinhwangmi’). Gemenet et al. found a positive effect from one allele of the *OR* gene [13]; given that there are only four copies, two of which are identical, it is likely that the *I. batatas* ‘Beauregard’ *OR* gene on chromosome 12D or 12F has a positive effect on β-carotene accumulation. This locus provides an opportunity to develop a genetic marker for orange flesh color for use in advanced breeding.

## 4. Materials and Methods

### 4.1. Plant Material

Previously, 779 sweet potato accessions in the USDA, ARS, Plant Genetic Resources Conservation Unit (PGRCU) germplasm repository from 7 geographic regions (Africa, Caribbean, Central America, South America, North America, East Asia, and Pacific Islands) including over 35 countries were genotyped and analyzed [9]. After removing accessions for quality control, 604 accessions remained with 150,293 SNPs used for analyses of genetic diversity and population structure [9]. From these analyses, and by employing a full-autopolyploid kinship matrix generated from allele dosage data [76], a 384-accession breeder subset that maintained genotypic diversity within a smaller, more manageable germplasm subset was created [9]. This 384-accession sweet potato germplasm subset was used in the GWAS analyses presented herein.

### 4.2. Storage Root Phenotypic Data

This study used phenotypic data of storage root dry matter content and color characteristics from the Germplasm Resources Information Network (GRIN) database and from a previous analysis of the USDA sweet potato germplasm collection [77]. The storage root traits include percent dry matter, subjective flesh (stele) color, subjective skin (periderm) color, flesh hue angle, and skin hue angle. Subjective flesh color and skin color were categorized using a numeric descriptor system as described [78]. For subjective flesh color, 1 = white, 2 = cream, 3 = dark cream, 4 = pale yellow, 5 = dark yellow, 6 = pale orange, 7 = intermediate orange, 8 = dark orange, and 9 = strongly pigmented with anthocyanins [78]. For subjective skin color, 1 = white, 2 = cream, 3 = yellow, 4 = orange, 5 = brownish orange, 6 = pink, 7 = red, 8 = purple-red, and 9 = dark purple [78]. The subjective color descriptions for each accession were from Jackson et al. [77]. Flesh and skin hue angle were obtained as described [77] using a tristimulus colorimeter, Konica–Minolta Chroma Meter (CR-400 with 8-mm aperture and 0° viewing angle; Konica-Minolta, Inc., Tokyo, Japan), within the CIE L*C*h color space. Each data point was an automatic average of three readings (8 mm diameter area) of the root surface (periderm) or flesh interior (stele). Accessions were sampled in quantities of either two (2012 and 2013) or three (2014) storage roots from each and either two (2012 and 2013) or one (2014) different locations on each root were measured with the colorimeter. Colorimeter calibration was performed using the manufacturer-provided standard white reference tile. Color Data Software CM-S100w SpectraMagic NX (Version 2.6) (Konica-Minolta 2014) was used to record data from the colorimeter. Percent dry weight (% dry matter) was determined for three roots for each of the two field replicates for each accession. From each root, a 1 cm long by 1 cm diameter transverse core was cut out, as described by Jackson et al. [77]. The cores were weighed (wet root weight) and dried at 80 °C for 48 h (AF Model 40 Lab Oven, Quincy Lab, Inc., Chicago, IL, USA) and then weighed again (dry root weight). In all, three data sets from [77] (periderm-field, stele-field, and percent dry weight-field) were used for GWAS analyses in this study.

### 4.3. DNA Isolation and Genotyping

Genotyping was completed as described previously [8]. DNA was isolated from lyophilized leaf tissue using the DNeasy Plant Mini Kit (Qiagen, Hilden, Germany). DNA integrity, purity, and concentration were determined via a NanoDrop 2000 spectrophotometer (ThermoFisher, Norristown, PA, USA). Genotyping was completed via a modified genotyping-by-sequencing method optimized for heterozygous and polyploid genomes GBSpoly [8,9]. Following DNA endonuclease double digestion (CviAII and TseI), barcode adapter oligos were ligated to DNA fragments. Secondary double digests (CviAII and TseI) were performed to remove chimeric sequences. The resulting pools were purified using the Blue Pippin Prep System (Sage Science, Beverly, MA, USA) with size-selection for 300–400 bp fragments. PCR amplification was repeated for 18 cycles (NEB Phusion high-fidelity polymerase; New England Biolabs, Ipswich, MA, USA), and these size-selected, amplified libraries were sequenced on an Illumina HiSeq 2500. Subsequent fastq data filtering was performed using the ngsComposer pipeline [79]. SNP dosage calling and variant filtering were performed using GBSapp (https://github.com/bodeolukolu/GBSapp, accessed on 28 October 2024), a bioinformatics pipeline that incorporates third-party tools into an original workflow to verify reproducibility of variant calling as described previously [8]. Variant filtering of SNPs used to generate files for this analysis used a minor allele frequency of (MAF) threshold of 0.02, read depth threshold of 45 for 6x dosages, and SNP and sample missing rate threshold of 0.3 and 0.3, respectively. Following filtering using an LD threshold of 0.1 and MAF SNP elimination, 47,423 SNPs remained for downstream analyses. Composite linkage disequilibrium (LD) was calculated using the ldsep v2.1.4 R Package (https://cran.rstudio.com/web/packages/ldsep/index.html, accessed on 28 October 2024; linkage disequilibrium shrinkage estimation for polyploids) [80]. LD was estimated for each pair of markers on the same chromosome, plotted, and generalized additive model smoothing was used to demonstrate rapid LD decay (Appendix A).

### 4.4. Genome-Wide Association Study Analyses

A genome-wide association study (GWAS) was performed employing the GWASpoly R package (https://jendelman.github.io/GWASpoly/GWASpoly.html, accessed on 28 October 2024) that is tailored for association studies in polyploid species [81]. To reduce the risk of false positives, kinship matrices were calculated to identify relatedness via the VanRaden method [82]. Seven genetic models were employed (additive, 1-dom-ref [1-dominant reference allele], 1-dom-alt [1-dominant alternate allele], 2-dom-ref, 2-dom-alt, 3-dom-ref, and 3-dom-alt) to capture the wide array of genetic influences on phenotypic expression. False discovery rate (FDR) corrections were used to determine significance of associations.

### 4.5. Identification of Candidate Genes

Following association analyses, highly significant SNPs were identified and investigated for links to candidate genes. SNPs were first identified using GBSpoly and the *Ipomoea trifida* and *Ipomoea triloba* reference genome assemblies (http://sweetpotato.uga.edu/). Next, BLAST searches of the *Ipomoea batatas* ‘Beauregard’ v1 reference genome assembly were employed to locate the identified SNPs within the *I. batatas* genome (http://sweetpotato.uga.edu/). Both *I. trifida* and *I. batatas* have 15 chromosomes, but as sweet potatoes are hexaploid, the *I. batatas* ‘Beauregard’ genome has six copies of each chromosome which are named A, B, C, D, E, and F. Thus, a single SNP in the *I. trifida* genome can map to up to six synonymous positions in the *I. batatas* ‘Beauregard’ genome, and structural chromosome changes such as gene duplications, amplifications, or deletions can lead to more or fewer gene copies, respectively. For both the *I. trifida* and the *I. batatas* ‘Beauregard’ genomes, any annotated genes within 100 kb of the SNPs were noted.

## 5. Conclusions

Translating the SNPs associated with sweet potato storage root traits identified by GWAS into a mechanistic understanding of the genetic control of starch and pigment accumulation is complicated. About half of the 28 SNPs identified here are located within intergenic regions, and most of the SNPs that do reside within genes are located within introns or 3′ UTRs (untranslated regions). Of the several SNPs that represent alleles that would result in proteins with different coding sequences, the function of the encoded proteins or their possible effect on the phenotypes investigated are unclear. Although the decision to search for target genes within a 200 kb target window was somewhat arbitrary, previous work suggests that this is a reasonable window [83] and this 200 kb was used in a similar study [48]. It is possible that an SNP could reside within an enhancer region that regulates a target gene outside the target windows studied here. Further, it is also probable that affects from more than one gene contribute to the phenotype associated with an individual SNP. Protein–protein interactions such as those between the OR and PSY or between OR and other proteins involved in plastid division could be affected by mutations in either partner. In addition, the node in the β-carotene metabolic pathway represented by the interaction between PSY and OR is just that—a component of a larger network. The steady-state level of β-carotene is affected by both the level of synthesis of carotenoid precursors in the MVA or MEP upstream of β-carotene synthesis, and by the level of degradation of β-carotene, for example, increased BCH activity. Nevertheless, one allele of OR does appear to function as a dominant gene for orange flesh color, and as mentioned above it should be possible to develop PCR-based markers, such as for use in dCAPS (derived cleaved amplified polymorphic sequence [84]), to screen for orange flesh color given the current state of knowledge. The genetic loci and candidate genes identified in this study provide targets to focus future efforts on designing similar genetic markers for dry matter content and flesh and skin color to use in marker-assisted selection of sweet potato varieties.

## Figures and Tables

**Figure 1 ijms-25-11727-f001:**
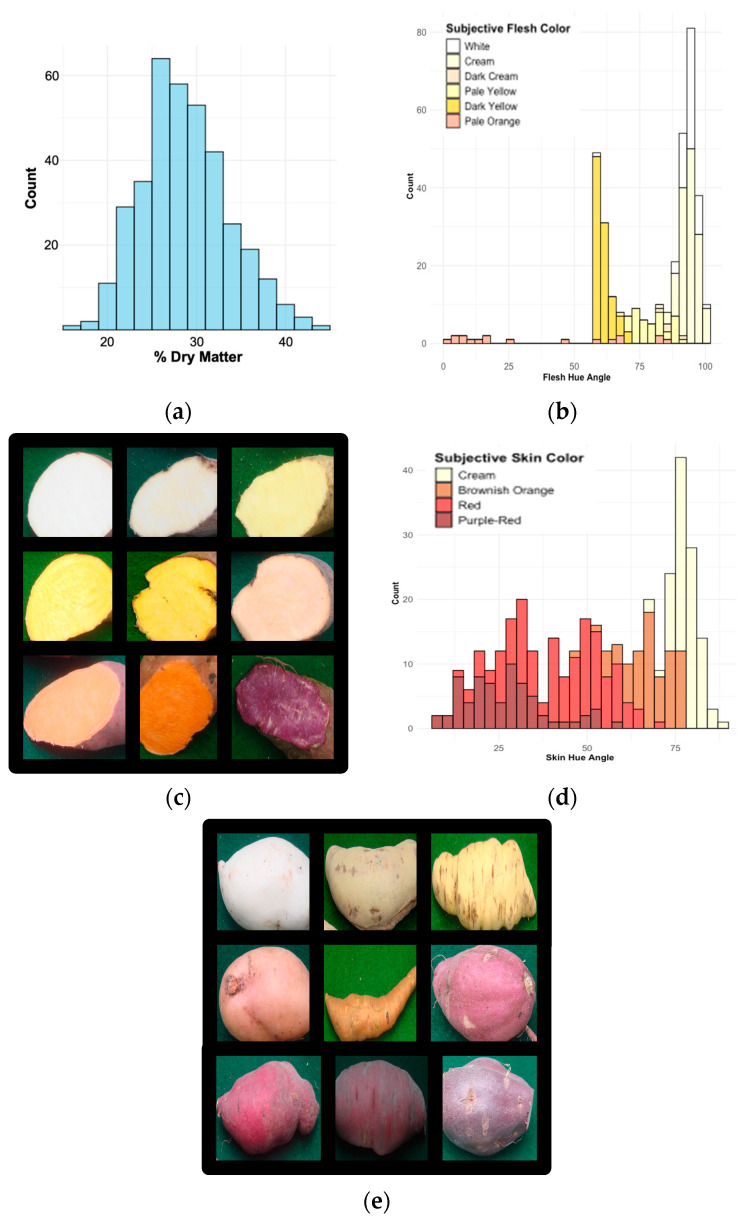
Distribution of sweet potato storage root traits. Phenotypes are represented as follows: (**a**) histogram of percent dry matter; (**b**) bar chart of flesh hue angle; (**c**) subjective flesh color where ratings are 1 = white (top left), 2 = cream (top middle), 3 = dark cream (top right), 4 = pale yellow (middle left), 5 = dark yellow (middle), 6 = pale orange (middle left), 7 = intermediate orange (bottom left), 8 = dark orange (bottom middle), and 9 = strongly pigmented with anthocyanins (bottom right) [17]; (**d**) histogram of skin hue angle; (**e**) subjective skin color where ratings are 1 = white (top left), 2 = cream (top middle), 3 = yellow (top right), 4 = orange (left middle), 5 = brownish orange (middle), 6 = pink (right middle), 7 = red (bottom left), 8 = purple-red (bottom middle), and 9 = dark purple (bottom right) [17].

**Figure 2 ijms-25-11727-f002:**
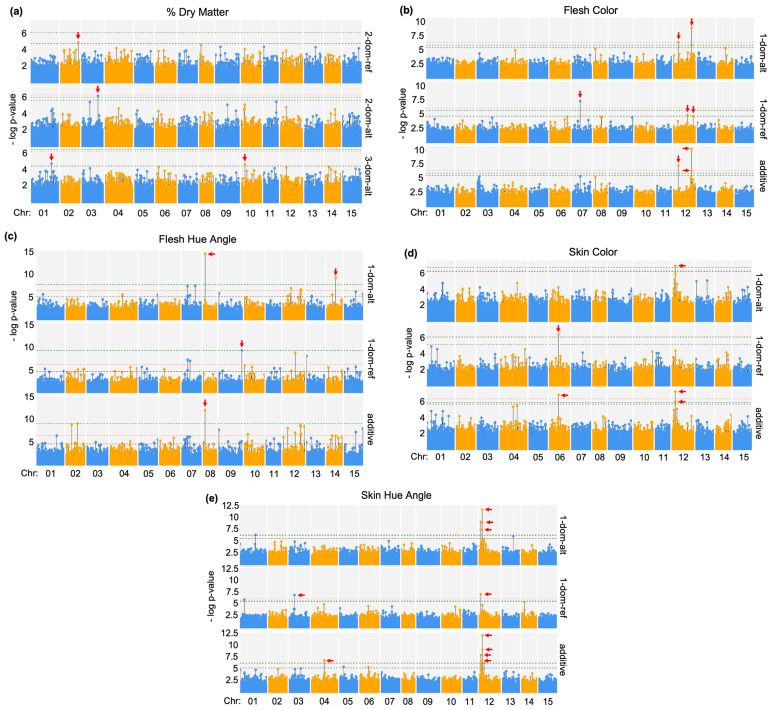
Manhattan plots of sweet potato storage root trait SNPs identified using GBSpoly GWAS analyses. Following GWAS analyses as described in the Materials and Methods, Manhattan plots were prepared for the traits of (**a**) percent dry matter, (**b**) flesh color, (**c**) flesh hue angle, (**d**) skin color, and (**e**) skin hue angle. For each trait, Manhattan plots displaying the most significant SNPs for each genetic model (1-dom-ref, 1-dom-alt, 2-dom-ref, 2-dom-alt, 3-dom-ref, 3-dom-alt, and additive) are illustrated.

**Figure 3 ijms-25-11727-f003:**
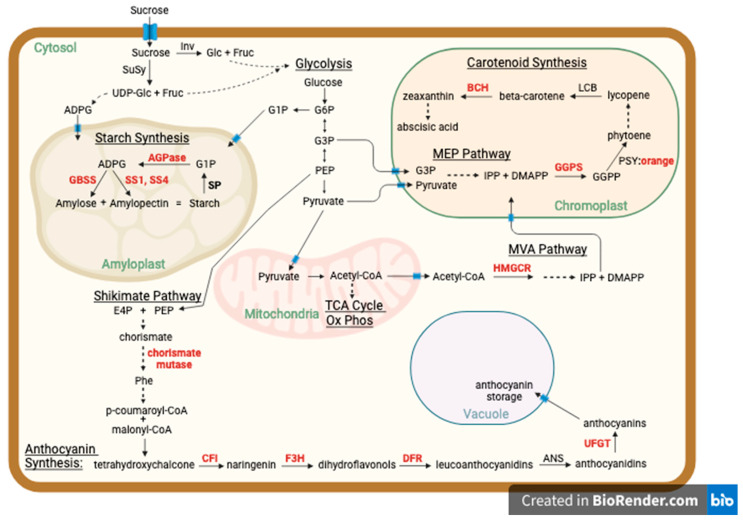
Connections of central carbon metabolism to starch, carotenoid, and anthocyanin biosynthesis with candidate metabolic enzymes from this study highlighted in red. Abbreviated core metabolic pathways are represented including glycolysis, the tricarboxylic acid (TCA) cycle, mevalonic acid (MVA) pathway, shikimate pathway, and the methylerythritol 4-phosphate (MEP) pathway. Sucrose from photosynthetic “source” tissue enters the sweet potato storage root cell. The disaccharide sucrose can be hydrolyzed to glucose (Glc) and fructose (Fruc) by invertase (*Inv*), and both Glc and Fruc can enter glycolysis. Alternately, sucrose synthase (*SuSy*) can convert sucrose to uridine-5-diphoshate (UDP)-glucose plus Fruc, which can enter glycolysis. UDP-glucose can be converted to adenosine-5-diphosphate-glucose (ADPG) that can enter the amyloplast (specialized starch-storing plastid) for starch synthesis. Alternately, ADPG can be synthesized from glucose 1-phosphate (G1P) inside the plastid by ADPG pyrophosphorylase (*AGPase*). Granule-bound starch synthases (*GBSS*) use ADPG to synthesize amylose, and soluble starch synthases (*SSs*) along with other enzymes synthesize amylopectin from ADPG. Starch phosphorylase (*SP*) is involved in starch breakdown. Carotenoid synthesis occurs in chromoplasts. The carotenoid precursors, isopentyl pyrophosphate (IPP), and dimethylallyl pyrophosphate (DMAPP), can be derived from the cytosolic MVA pathway, which uses acetyl CoA (acetyl Coenzyme A) as a precursor with the rate-limiting step being catalyzed by HMG CoA reductase (*HMGCR*), or IPP and DMAPP can be derived from the plastid MEP pathway, which uses the glycolytic intermediates glyceraldehyde 3-phosphate (G3P) and pyruvate as precursors. Geranylgeranyl pyrophosphate synthase (*GGPS*) synthesizes geranylgeranyl pyrophosphate (GGPP) from IPP and DMAPP. Phytoene is synthesized by phytoene synthase (*PSY*), which is stabilized by the *ORANGE* protein. Beta-carotene hydroxylase (*BCH*) oxidizes β-carotene, thereby decreasing β-carotene levels. The isopropanoid pathway for anthocyanin biosynthesis is also illustrated. The glycolytic intermediate phosphoenolpyruvate (PEP) and erythrose 4-phosphate (E4P) from the pentose phosphate cycle (not illustrated) can enter the shikimate pathway wherein chorismate mutase catalyzes the rate-limiting step and the flavonoid precursor phenylalanine (Phe) is synthesized. Chalcone-flavanone isomerase (*CFI*), flavanone 3-hydroxylase (*F3H*), dihydroflavonol reductase (DFR), and UDP-glucose flavonoid 3-O-glucosyltransferase (*UFGT*) act consecutively in the synthesis of anthocyanins, which are stored in vacuoles. Abbreviations: ADPG—ADP-glucose; AGPase—ADP-glucose pyrophosphorylase; BCH—beta-carotene hydroxylase; CFI—chalcone-flavanone isomerase; DFR—dihydroflavonol reductase; DMAPP—dimethylallyl pyrophosphate; E4P—erythrose 4-phosphate; F3H—flavanone 3-hydroxylase; G1P—glucose 1-phosphate; G6P—glucose 6-phosphate; G3P—glyceraldehyde 3-phosphate; GBSS—granule-bound starch synthase; GGPP—geranylgeranyl pyrophosphate; GGPS—geranylgeranyl pyrophosphate synthase; HMGCR—hydroxy methyl glutarate CoA reductase; IPP—isopentenyl pyrophosphate; MEP—methylerythritol 4-phosphate; MVA—mevalonic acid pathway; PEP—phosphoenolpyruvate; Phe—phenylalanine; PSY—phytoene synthase; SP—starch phosphorylase; SS—starch synthase; TCA—tricarboxylic acid cycle; UGFT—UDP-glucose flavonoid 3-O-glucosyltransferase.

**Table 1 ijms-25-11727-t001:** Name and *I. trifida* position of SNPs associated with storage root traits. SNPs are listed by traits of dry matter (DM), flesh color (FC), flesh hue angle (FHA), skin color (SC), and skin hue angle (SHA). For each trait, SNPs are numbered consecutively to give a unique SNP name, and the positions in the *I. trifida* reference genome are indicated (http://sweetpotato.uga.edu/ accessed on 4 September 2024).

Trait	SNP Name	Genomic Position
Dry Matter	DM1	Chr01: 27,238,168
	DM2	Chr02: 23,017,613
	DM3*	Chr03: 18,888,440 *
	DM4	Chr10: 3,229,129
Flesh Color	FC1	Chr07: 5,556,076
	FC2	Chr12: 3,933,566
	FC3	Chr12: 18,956,680
	FC4	Chr12: 22,042,439
	FC5	Chr12: 22,064,285
	FC6	Chr12: 23,146,784
Flesh Hue Angle	FHA1	Chr08: 1,457,982
	FHA2	Chr09: 23,213,578
	FHA3	Chr14: 11,541,393
Skin Color and Hue Angle	SC1	Chr06: 18,709,220
	SC2, SHA1	Chr12: 2,823,583
	SC3, SHA2	Chr12: 2,841,327
	SHA3	Chr01: 3,487,779
	SHA4	Chr01: 23,116,643
	SHA5	Chr03: 5,991,712
	SHA6	Chr04: 12,982,060
	SHA7*	Chr12: 1,568,751 *
	SHA8*	Chr12: 1,854,779 *
	SHA9	Chr12: 4,042,610

* Note: The DM3* SNP comprises two SNPs that are 12 bp apart, the SHA7* SNP comprises four SNPs within 5 bp of each other, and the SHA8* SNP comprises two SNPs that are 3 bp apart.

**Table 2 ijms-25-11727-t002:** Transmembrane transporter genes near SNPs associated with storage root traits. Candidate transmembrane transporter genes within or near SNPs identified in this study are listed including: the *I. batatas* ‘Beauregard’ v1 id and annotation (http://sweetpotato.uga.edu/), homologous genes, and references providing functional information.

SNP	Gene ID	Annotation	Homologs	References into Function
DM3*	03E_G012960	unknown function DUF914	SLC35	solute carrier family 35 nucleotide sugar transporters, drug metabolite transporter family [23]
DM3*	03D_G014090	H^+—^ATPase	ATPase 4	primary active transport; overexpression increases rice yield [56]
FC5	12C_G029170	nucleotide-diphospho-sugar transferase family	unknown	predicted, putative ADP-glucose transporter
SHA9	12C_G008210	phosphoenolpyruvate (PEP)/phosphate translocator	PPT2	PEP/Pi Translocator, PEP transport for shikimate pathway [57]

**Table 3 ijms-25-11727-t003:** Transcription factor genes near SNPs associated with storage root traits. Candidate transcription factor genes within or near SNPs identified in this study are listed including: the *I. batatas* ‘Beauregard’ v1 id and annotation (http://sweetpotato.uga.edu/), homologous genes, and references providing functional information.

SNP	Gene ID	Annotation	Homologs	References into Function
DM4	10A_G004370	bHLH	bHLH 144	starch synthesis in rice [60]
FC3	12D_G020790	bHLH	bHLH 6	anthocyanin biosynthesis in carrot [28]
FC6	12E_G025410	bHLH	SPEECHLESS	stomatal lineage development, stemness [61]
FHA1	08F_G003550	bHLH	bHLH 94	development/embryogenesis [36]
SC2, SHA1	12D_G004200	bHLH	SPATULA	root meristem development [62]
SHA5	03F_G010380	bHLH	bHLH 18	jasmonic acid regulated Fe^2+^ transport [63]
FC1	07A_G006800	Myb	Myb 62	negative regulator of anthocyanin biosynthesis [25]
FC2	12C_G007940	Myb	Myb 16	positive regulator cutin and wax biosynthetic genes [26]
SHA9	12B_G007380	Myb	Myb 16	
FC4, FC5	12C_G029230	Myb	Myb 4	negative regulator of anthocyanin biosynthesis [31]
SHA5	03B_G012820	Myb	Myb 44	salicylic and jasmonic acid signaling, WRKY70 regulation [64]
SHA8*	12E_G003940	Myb	RAX3	meristem development [65]
SHA9	12C_G008030	Myb-family	DMTF1	cyclin D binding Myb Transcription Factor 1;, aka DMP1 [66]
DM1	01F_G025330	WD40 repeat	PTP1	periodic tryptophan protein 1, chromatin-associated, ribosome biogenesis (growth) [67]
SHA7	12C_G004320	WD40 repeat	APRF1	anthesis promoting factor 1, epigenetic regulation of flowering time, HSP90 interaction [46]
SHA9	12C_G008110	WD40 repeat	WDR12	ribosome biogenesis (growth), high expression in roots [68]
DM2	02B_G028270	WRKY	WRKY 40	transcriptional repressor of abscisic acid signaling [22]
DM5	10C_G004950	WRKY	WRKY 71	meristem development [24]
FC4, FC5	12F_G025420	WRKY	WRKY 70	integration of salicylic and jasmonic acid signaling, Myb-44 regulates [30]
FC4, FC5	12E_G024250	WRKY	WRKY 55	salicylic acid and ROS signaling [69]
SHA5	03B_G012780	WRKY	WRKY 7	salicylic acid responsive, regulation of flavonoid metabolism [70]
SHA4	01A_G021900	Redox responsive	ERF109	jasmonic acid, auxin signaling [71]
FHA1	08A_G002660	jumonji	JMJ706	chromatin remodeling H3K9 demethylase, rice flower development [34]
SHA3	01A_G008090	homeobox-1	RLT1	chromatin remodeling, jasmonic acid biosynthesis [72]
FC3	12B_G022290	Phytochrome-associated	IAA27-like	auxin signaling, root development [27]

## Data Availability

The data sets presented in this study can be found in online repositories. The names of the repository/repositories and accession number(s) can be found below: https://www.ncbi.nlm.nih.gov/, PRJNA880973, accessed on 28 October 2024.

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
