# Peer review of "Genome-Wide Association Study of Sweet Potato Storage Root Traits Using GWASpoly, a Gene Dosage-Sensitive Model"

_ijms, 2024, doi:10.3390/ijms252111727_

Round 1

Reviewer 1 Report

Comments and Suggestions for Authors

The article, by Robert R. Bowers and colleagues, deals with the genetics of Ipomoea batatas. In terms of subject matter, the article is consistent with the IJMS theme. The article is well structured. Linguistically correct. In the introduction, the authors comprehensively discuss the topic. The molecular methods used are correct. The authors' highlights include: - identification of genes responsible for metabolism of sweetpotato storage root starch, carotenoids, and anthocyanins; - identification of transmembrane transporters affect sweetpotato metabolism.

Congratulations to the authors for a solid job.

Minor comments: currently we write the names of families without italics, please correct.

in “materials” please give the full Latin name of the studied species

Author Response

Reviewer #1

The article, by Robert R. Bowers and colleagues, deals with the genetics of Ipomoea batatas. In terms of subject matter, the article is consistent with the IJMS theme. The article is well structured. Linguistically correct. In the introduction, the authors comprehensively discuss the topic. The molecular methods used are correct. The authors' highlights include: - identification of genes responsible for metabolism of sweetpotato storage root starch, carotenoids, and anthocyanins; - identification of transmembrane transporters affect sweetpotato metabolism.

Congratulations to the authors for a solid job.

Minor comments: currently we write the names of families without italics, please correct.

in “materials” please give the full Latin name of the studied species

Dear Reviewer #1:

Thank you for the kind words. The italics in the family name was removed; it now reads "Convolvulaceae".  In "materials" the full Latin name including the genus Ipomoea is now included. Thank you for pointing out these changes.

Reviewer 2 Report

Comments and Suggestions for Authors

The paper is very descriptive. Represents a high amount of work and novel data, but as the authors state in the conclusion, the results are very descriptive and the significance is quite ambigous, as there are no cause-effect data, given the difficult to check whether the identified SNP are significative or not. Balancing all these factors my opinion is that the paper merits publication as the amount of novel information may be very useful for breeders and for crop biotechnology.

I have some minor comments:

Figure 3: Please highlight the relevant enzymes with a different color, as using bold font is not enough to distinguish the information obatained in this report from the previous information. In adition the figure legend is very narrative, and seems part of the discussion. Please in the figure legend only describe what is represented in the figure, and the relevant information include in the discussion, citing the figure.

Results: are too arid and long, describing gene by gene. Can you summarize this information in a table per section and only refer to the most relevant data or the closes SNP to the coding region?

Discussion: I have missed considering the biotech improvement made in sweet potato (cripsr or GMO). Is any of the published modification close to any of the identified SNP, it could be potato or any related crop. Is there any mutant described in sweet potato affecting any of the considered SNP. This information is not present or scatterd in different sections, I would put all together as it is relevant to validate the results.

Author Response

Reviewer #2

The paper is very descriptive. Represents a high amount of work and novel data, but as the authors state in the conclusion, the results are very descriptive and the significance is quite ambiguous, as there are no cause-effect data, given the difficult to check whether the identified SNP are significative or not. Balancing all these factors the paper merits publication as the amount of novel information may be very useful for breeders and for crop biotechnology.

I have some minor comments:

Figure 3: Please highlight the relevant enzymes with a different color, as using bold font is not enough to distinguish the information obtained in this report from the previous information. In addition, the figure legend is very narrative and seems part of the discussion. Please in the figure legend only describe what is represented in the figure, and the relevant information include in the discussion, citing the figure.

Results: are too arid and long, describing gene by gene. Can you summarize this information in a table per section and only refer to the most relevant data or the closes SNP to the coding region?

Discussion: I have missed considering the biotech improvement made in sweet potato (cripsr or GMO). Is any of the published modification close to any of the identified SNP, it could be potato or any related crop. Is there any mutant described in sweet potato affecting any of the considered SNP. This information is not present or scattered in different sections, I would put all together as it is relevant to validate the results.

Dear Reviewer #2:

Thank you for your review of and comments on the paper. The following changes were made in response to the "minor comments."

In Figure 3, the relevant enzymes are now highlighted in red as suggested. In addition, the figure legend of Figure 3 was shortened as suggested.

The Results are admittedly dry; the intention was to present each identified SNP along with the closest and most relevant genes. This study presents a great deal of information, as you note, and great effort was exerted to present the information as clearly and concisely possible. The Results are summarized in Table S7 where a total of 82 genes are listed. As we identified 28 SNPs, this is slightly less than 3 genes / SNP.

Discussion. I think you meant to say identified gene...considered gene rather than "identified SNP...considered SNP." There are some publications from researchers in China that use Crispr or other genetic manipulations to alter some of the genes considered in the current paper. In response to your comment, the following sentence was added to the discussion " Studies have demonstrated that genetic modification of these starch biosynthetic enzymes can alter starch metabolism in sweetpotatoes [1, 2]

  1. Wang Y, Li Y, Zhang H, Zhai H, Liu Q, He S: A soluble starch synthase I gene, IbSSI, alters the content, composition, granule size and structure of starch in transgenic sweet potato. Sci Rep 2017, 7(1):2315.
  2. Wang H, Wu Y, Zhang Y, Yang J, Fan W, Zhang H, Zhao S, Yuan L, Zhang P: CRISPR/Cas9-Based Mutagenesis of Starch Biosynthetic Genes in Sweet Potato (Ipomoea Batatas) for the Improvement of Starch Quality. Int J Mol Sci 2019, 20(19).

As noted in the discussion under Sweetpotato Genomic Resources, it is difficult to compare genes identified in different studies that utilize different genomic resources, but it does appear that "IbSS1" from references 52 and 53 refer to the same SS1 gene as our study.

Thank you again for the review. We think the manuscript is improved by the changes made in response.

Reviewer 3 Report

Comments and Suggestions for Authors

This is important work identifying candidate genes for nutritional traits in sweet potato, to be validated in future studies. I can’t comment on the metabolic pathways described or whether the sweet potato literature has been adequately cited, and instead focus on the GWAS methodology.

Fig. S3C, Tables S2, S3, S5, S6, and Lines 132-138: Given the low LD shown in Fig. S2, 100 kb seems to be far too high of a cutoff for finding candidate genes that may be linked to the significant SNPs.  I recommend 25 kb or maybe even less (see my comment on Fig. S1 for making it easier to determine a good threshold). It seems very unlikely that two loci 100 kb apart would be in linkage disequilibrium. This also brings into question whether the ORANGE gene was truly detected in this study (lines 462-463). The citation of Brodie et al. (2016; line 753) suggests that perhaps many significant SNPs are in or near enhancers that are very distant from the affected genes, although I would interpret that study with caution given that it was performed using human data.  If the authors feel strongly about not shortening their list of candidate genes, I would like them to at least distinguish between those within LD of significant SNPs and those within potential enhancer distance of significant SNPs.

Fig. S1 would be easier to interpret if the points were partially transparent.  For example in ggplot2 set geom_point(alpha = 0.2) or something similar.

Fig. S3: Parts A and B are difficult to interpret from the figure caption alone.

Lines 102-109: What is the LD between pairs of significant SNPs that are physically close to each other? Do they seem to all be linked to the same functional variant, or multiple variants likely to impact the same gene?

I noticed in the discussion and methods that there are many places where “I. batatas” and “I. trifida” are not italicized.

The Q-Q plots in Fig. S2 look good, so I think kinship has been adequately controlled for.

Lines 727-730: Bonferroni is always more stringent than FDR, so I don’t understand how both were used for determining significance.  The P = alpha/n that is described sounds like Bonferroni.

Author Response

Reviewer #3

This is important work identifying candidate genes for nutritional traits in sweet potato, to be validated in future studies. I can’t comment on the metabolic pathways described or whether the sweet potato literature has been adequately cited, and instead focus on the GWAS methodology.

Dear Reviewer #3:

Thank you for your review of and comments on the paper.

Fig. S3C, Tables S2, S3, S5, S6, and Lines 132-138: Given the low LD shown in Fig. S2, 100 kb seems to be far too high of a cutoff for finding candidate genes that may be linked to the significant SNPs.  I recommend 25 kb or maybe even less (see my comment on Fig. S1 for making it easier to determine a good threshold). It seems very unlikely that two loci 100 kb apart would be in linkage disequilibrium. This also brings into question whether the ORANGE gene was truly detected in this study (lines 462-463). The citation of Brodie et al. (2016; line 753) suggests that perhaps many significant SNPs are in or near enhancers that are very distant from the affected genes, although I would interpret that study with caution given that it was performed using human data.  If the authors feel strongly about not shortening their list of candidate genes, I would like them to at least distinguish between those within LD of significant SNPs and those within potential enhancer distance of significant SNPs.

Response:

Calculations of linkage disequilibrium are notoriously difficult, especially in polyploid plants. The rapid decay of LD illustrated in figure S1 demonstrates that most of the SNPs in the GWAS analysis behave independently. The calculations of LD assume that there is sufficient information contained in pairs of isolated markers to detect recombination between them [3]. Due to the complex polyploid sweetpotato genome and the limited population sizes analyzed, however, this assumption is not valid. Additional uncertainty arises from the fact that the SNPs identified through the genotyping-by-sequencing method herein are mapped to the ancestral I. trifida and I. triloba genomes.

As noted in the paper, the decision to search for potential candidate genes within 100 kb of the SNPs was somewhat arbitrary. Throughout the ~25-year history of GWAS, assigning causative genes to SNPs has been difficult. In most instances, including in the present study, SNPs are located in intergenic regions or introns - thus, there is no obvious functional consequence of the SNPs. Further, many of the SNPs identified by GWAS are not in LD with any gene [4]. For the SNP to be a true positive result, however, the SNP must be genetically linked to the causative gene. Although many studies have focused efforts on the gene closest to the SNP, several authors have demonstrated that using windows of up to 500 kb have enabled many more causative genes to be identified.

There appears to be a common misunderstanding of genetic distance (i.e. what constitutes a short distance) in the literature. In humans, two genes that are 1 centimorgan (1% recombination rate) apart are about 1,000,000 base pairs apart. Two genes 100 kb apart are tightly genetically linked. In plants and especially polyploid plants recombination frequencies may be higher, but two genes 100 kb apart would still be expected to be genetically linked based on classical molecular genetics. The average physical distance per centimorgan is observed to be 244 kb for the rice genome (https://www.ncbi.nlm.nih.gov/pmc/articles/PMC150577/) and ~150 kb for sweetpotato genome [3]. The averaged genetic to physical map ratios for were 124.8 kb per cM for I. trifida and 114.9 kb per cM for I. triloba in ref 3.

Enhancers and repressors typically function over distances greater than 500 kb. We are not suggesting that we are identifying these genetic elements (or consequences of them) herein.

Fig. S1 would be easier to interpret if the points were partially transparent.  For example in ggplot2 set geom_point(alpha = 0.2) or something similar.

Response:

Unfortunately, we were unable to re-run the analyses to regenerate these plots in the limited editing period.

Fig. S3: Parts A and B are difficult to interpret from the figure caption alone.

Response:

The figure caption was edited.

Lines 102-109: What is the LD between pairs of significant SNPs that are physically close to each other? Do they seem to all be linked to the same functional variant, or multiple variants likely to impact the same gene?

Response:

Please see response to the first point above. Although LD may not be detected between the significant SNPs that are physically close to each other - for example FC4 and FC5, which are ~ 22 kb apart - it seems very likely to the authors that (a) these two SNPs are genetically linked and (b) the orange gene, which is about 70 kb from these SNPs is at least one of the causative genes responsible for both of these significant SNPs. In addition, for each SNP it is quite possible that more than one gene contributes to the complex phenotypes investigated. There is a chalcone flavanone isomerase gene (encodes a key, rate-limiting enzyme in anthocyanin synthesis) and a Myb4 gene (transcriptional regulator of anthocyanin synthesis genes) near the orange gene (key determinant of beta-carotene accumulation). Various combinations of different alleles of each of these genes could contribute to the relative accumulation of orange and purple pigments in sweetpotatoes.

I noticed in the discussion and methods that there are many places where “I. batatas” and “I. trifida” are not italicized.

Response:

Thank you for pointing this out. Something happened during the editing process which is now corrected.

The Q-Q plots in Fig. S2 look good, so I think kinship has been adequately controlled for.

Response:

Thank you for noting this.

Lines 727-730: Bonferroni is always more stringent than FDR, so I don’t understand how both were used for determining significance.  The P = alpha/n that is described sounds like Bonferroni.

Response:

The False Discovery Rate (FDR) to enable identify as many candidate genes as possible. About half of the identified SNPs are significant with the more stringent Bonferroni.

  1. Wang Y, Li Y, Zhang H, Zhai H, Liu Q, He S: A soluble starch synthase I gene, IbSSI, alters the content, composition, granule size and structure of starch in transgenic sweet potato. Sci Rep 2017, 7(1):2315.
  2. Wang H, Wu Y, Zhang Y, Yang J, Fan W, Zhang H, Zhao S, Yuan L, Zhang P: CRISPR/Cas9-Based Mutagenesis of Starch Biosynthetic Genes in Sweet Potato (Ipomoea Batatas) for the Improvement of Starch Quality. Int J Mol Sci 2019, 20(19).
  3. Mollinari M, Olukolu BA, Pereira GdS, Khan A, Gemenet D, Yencho GC, Zeng Z-B: Unraveling the Hexaploid Sweetpotato Inheritance Using Ultra-Dense Multilocus Mapping. G3 Genes|Genomes|Genetics 2020, 10(1):281-292.
  4. Brodie A, Azaria JR, Ofran Y: How far from the SNP may the causative genes be? Nucleic Acids Res 2016, 44(13):6046-6054.